# Genome-Wide Association Studies of Seven Root Traits in Soybean (*Glycine max* L.) Landraces

**DOI:** 10.3390/ijms24010873

**Published:** 2023-01-03

**Authors:** Seong-Hoon Kim, Rupesh Tayade, Byeong-Hee Kang, Bum-Soo Hahn, Bo-Keun Ha, Yoon-Ha Kim

**Affiliations:** 1National Agrobiodiversity Center, National Institute of Agricultural Sciences, RDA, Jeonju 5487, Republic of Korea; 2Department of Applied Plant Science, Chonnam National University, Gwangju 61186, Republic of Korea; 3Department of Applied Biosciences, Kyungpook National University, Daegu 41566, Republic of Korea

**Keywords:** soybean, root trait, GWAS, SNP, candidate gene

## Abstract

Soybean [*Glycine max* (L.) Merr.], an important oilseed crop, is a low-cost source of protein and oil. In Southeast Asia and Africa, soybeans are widely cultivated for use as traditional food and feed and industrial purposes. Given the ongoing changes in global climate, developing crops that are resistant to climatic extremes and produce viable yields under predicted climatic conditions will be essential in the coming decades. To develop such crops, it will be necessary to gain a thorough understanding of the genetic basis of agronomic and plant root traits. As plant roots generally lie beneath the soil surface, detailed observations and phenotyping throughout plant development present several challenges, and thus the associated traits have tended to be ignored in genomics studies. In this study, we phenotyped 357 soybean landraces at the early vegetative (V2) growth stages and used a 180 K single-nucleotide polymorphism (SNP) soybean array in a genome-wide association study (GWAS) conducted to determine the phenotypic relationships among root traits, elucidate the genetic bases, and identify significant SNPs associated with root trait-controlling genomic regions/loci. A total of 112 significant SNP loci/regions were detected for seven root traits, and we identified 55 putative candidate genes considered to be the most promising. Our findings in this study indicate that a combined approach based on SNP array and GWAS analyses can be applied to unravel the genetic basis of complex root traits in soybean, and may provide an alternative high-resolution marker strategy to traditional bi-parental mapping. In addition, the identified SNPs, candidate genes, and diverse variations in the root traits of soybean landraces will serve as a valuable basis for further application in genetic studies and the breeding of climate-resilient soybeans characterized by improved root traits.

## 1. Introduction

Soybean [*Glycine max* (L.) Merr.] is an important oilseed crop and an inexpensive source of protein and oil. In regions of Southeast Asia and Africa, soybeans are commonly used as a source of traditional food and animal feed and for diverse industrial purposes. In addition to these uses, given its symbiotic associations with nitrogen-fixing rhizobacteria, cultivating soybean crops can make a valuable contribution to fixing atmospheric nitrogen, thereby reducing reliance on the excessive application of chemical fertilizers and thus facilitating sustainable cultivation [1,2].

To meet the growing demand for food that will be necessary to sustain human populations in the coming years, it will be necessary to boost productivity. However, given the current yields of major crops, it will be difficult to meet the projected future demand. In addition to population growth, the predicted changes in global climate will potentially have wide-ranging ramifications with respect to human health, world trade flow, food prices, and the ecosystem, and could also imperil global food security [3,4]. Consequently, it is deemed imperative to make efforts to enhance the yields of soybean for food production. According to recent reports, compared with the previous growth season, global soybean production in 2021/2022 will be reduced by 9.2 million tons to 372.56 million metric tons, owing to the lower yields of South American crops [5]. This reduction in yield has been influenced by several factors, including soil properties and biotic and abiotic stresses. Typically, growth and development are detrimentally affected when plants are subject to stress, which is ultimately manifested in declining yields. To overcome the challenges presented by such yield reductions, a multi-faceted research approach is required. Generally, it is considered that roots serve as a frontline defensive barrier under conditions in which plants are exposed to stress [6]. From a morpho-physiological standpoint, roots play essential roles in water and nutrient uptake, transportation, and absorption (Freschet et al., 2021). Accordingly, the structure and function of roots are of particular importance in terms of crop productivity, and not surprisingly, root research is assuming increasing prominence in agricultural sectors to increase productivity from a broad perspective. At present, however, compared with other agronomical traits, agriculture-related plant root research has been comparatively limited, as root phenotyping tends to be tedious, expensive, and technically difficult, and thus to date has made little contribution to furthering crop improvement. However, based on recent technological developments, researchers have begun to use different approaches for phenotyping the roots of diverse plant species [7,8,9,10,11].

Typically, it is considered that the landraces of cultivated crop plant species, including soybean, are characterized by wide genetic diversity and carry multiple favorable alleles and genes that confer greater adaptability to adverse conditions than that shown by modern cultivars [12,13,14]. Consequently, conducting genomics studies using landraces is viewed as a particularly fruitful approach for assessing the available diversity for targeted traits. A plant’s resilience to biotic and abiotic stresses can also be improved by modification in the root. Root traits [root system architecture (RSA) and root morphological traits], tend to be quantitative in nature. Mainly, RSA comprises multiple traits, for example, total root length, root diameter, root biomass, root volume, etc., and can be used to assess a plant’s capacity to adapt to challenging conditions. These traits are influenced by multiple factors, and thus identifying quantitative trait loci (QTLs) associated with root traits and the associated molecular markers will represent a key approach for the exploitation of root traits in future breeding programs and crop improvement.

The increasingly widely adopted genome-wide association studies (GWASs) have facilitated the discovery of the immense allelic diversity harbored within natural populations with both high resolution and accuracy, and have thereby enabled the identification of genomic regions/QTLs linked to desirable traits. Using high-density SNP markers, researchers have succeeded in discovering genomics regions, loci, or alleles for root traits in numerous plant species, including soybean [15,16,17,18]. As a consequence of the discovery of heritable natural variations in soybean root traits, several QTLs associated with RSA have been identified using a mapping approach [16,19,20]. In addition, Ye et al. (2018) identified a major QTL (*qWT_Gm03)* linked to waterlogging tolerance and the regulation of RSA and root plasticity in soybean. Furthermore, based on a combined approach using genotyping-by-sequencing in conjunction with whole-genome sequencing, Seck et al. (2020) recently identified 10 QTL regions associated with total root length and primary root diameter in soybean [21]. All these studies indicated the significance of RSA in root development.

Apart from the markers identified in these studies, however, there has generally been a lack of relevant markers for root traits in molecular breeding, and few underlying genes have been identified for RSA [16,20,22,23]. Moreover, the requisite functional characterization or cloning of genes has yet to be performed. Thus, there currently exists a paucity of relevant information, particularly with respect to the identification of genomic regions or loci and alleles associated with biological progression and the effects on root traits. This scarcity of functional genomic information for root traits accordingly motivates identifying a more extensive range of unique genomic regions and the associated candidate genes necessary for elucidating the genetic basis of root traits of interest.

In this study, we aimed to examine the phenotypic association among root traits, elucidate their genetic basis, and identify significant SNPs associated with regions/loci controlling root traits. To this end, we phenotyped 357 soybean landraces at the early vegetative (V2) stage of growth and conducted GWAS using a soybean 180K SNP array. This study will help us comprehend the genetic basis of soybean root development.

## 2. Results

### 2.1. Root Trait Variation

In this study, 357 soybean germplasms were analyzed for seven root traits, all seven of which showed significant differences among the landraces (Appendix A). The range of the average root diameter (DIAM) was found to have a relatively even distribution across the germplasms, ranging in value from 0.404 to 0.666 mm, with a mean diameter of 0.537 mm. With respect to LAD and LAL, we obtained values ranging from 0.429 to 0.725 and 0.109 to 0.379, respectively, with corresponding mean values of 0.5637 and 0.216. The total root length (LENGTH) of the landraces ranged from 703.33 to 3212.642 cm, with a mean value of 1766.26 cm (Appendix A), whereas values obtained for the number of forks (NF) ranged from 1180.00 to 11,941.67, with a mean value of 5250.51. A majority of the germplasms had fork numbers ranging from 3200 to 5200, with numbers gradually declining at the higher and lower extremes. Assessment of root tip number (NT) revealed values distributed across a range from 6222.67 to 3255.00, with a mean number of 1777.58, whereas surface area (SA) values ranged from 124.15 to 491.667 cm^2^, with a mean value of 294.382 cm^2^ (Appendix A). Based on the determinations of skewness and kurtosis, we established all seven traits (DIAM, LAL, LAD, LENGTH, NT, NF, and SA) to be normally distributed (Table 1, Appendix A). Among these traits, the highest coefficient of variation (CV) was obtained for NF (32.23%), followed by NT (29.53%), LENGTH (23.92%), SA (22.50%), LAL (15.12%), LAD (9.25%), and DIAM (8.68%). These findings indicate a notable variation in the levels of significant difference among the measured traits.

### 2.2. Population Structure and Principal Component and Phylogenetic Analyses

Grouping the 357 soybean landraces into clusters corresponding to their major categories is a key step in highlighting the basic population structure of these landraces. Although genetic studies of landraces can reveal distinct genotypes, direct and precise genotype partitioning is difficult. As a result, we used K-means clustering algorithms to identify and group these landraces based on their similarities. We used the elbow method, which enables the determination of the number of clusters by the curve’s elbow (Appendix A). Using the elbow approach, we detected three clusters/groups (i.e., K = 3), with groups I, II, and III containing 92, 105, and 160 landraces, respectively, all three of which were found to have varying degrees of admixture. Landraces from South Korea were more or less equally distributed between groups I and III, with 51 and 59 landraces, respectively, whereas 71 of the 90 landraces from China were clustered in group II. North Korean landraces were found to be distributed within all three groups, with groups I, II, and III containing 22, 16, and 20 landraces, respectively. Similarly, Japanese landraces had representatives in each of the three groups with 11, 13, and 70 clustering in groups I, II, and III, respectively. Overall, we found that except for five landraces, the soybean landraces of South Korean origin could mainly be divided into two groups, whereas the majority of landraces originating from China fall into group II, and a mixed type of separation was observed for landraces of North Korean origin among three groups, without any distinct origin-based separation.

By way of confirming the population structure of the assessed soybean landraces, we performed phylogenetic and principal component analyses (PCA). For phylogenetic tree construction, we used all 357 landraces, which were compared based on genetic similarity matrix information. The results of both PCA and phylogenetic analysis were consistent with the aforementioned population structure (Figure 1). Phylogenetic analysis partitioned the 357 landraces into three major groups (Figure 1B), among which group I comprised 143 landraces [South Korea (13), Japan (27), North Korea (27), and China (82)], group II comprised 114 landraces [South Korea (45), Japan (59), North Korea (8), and China (2)], and group III comprised 100 landraces [South Korea (57), Japan (14), North Korea (23), and China (6)]. Similarly, PCA-based clustering revealed that the 357 soybean lines can be divided into three major groups, among which there was a certain degree of overlap comprising areas of admixture (Figure 1A). PC1 and PC2 accounted for 34.23% and 14.10% of the observed variation, respectively (Figure 1C). The first three, PC1, PC2, and PC3, contributed more than fifty percent (58.20%) variation (Appendix A). Based on these analyses, we can thus conclude that the 357 assessed soybean landraces can be divided into three major groups/subpopulations, which represent an admixture of three ancestral populations. 

### 2.3. Estimation of Linkage Disequilibrium Decay

To gain an estimate of LD decay, we performed pairwise comparisons between all filtered SNPs. At a cut-off value of *r*^2^ = 0.1, the average LD decay distance of the 357 soybean landraces was approximately 200 kb, *r^2^* = 0.2 (Figure 1D). Furthermore, we determine the pattern of LD across the genome to establish the number of haplotype blocks containing SNPs that can be used to determine the candidate gene identification range.

### 2.4. Genome-Wide Association Analysis

To identify SNPs that are significantly associated with the seven assessed root traits (DIAM, LAD, LAL, LENGTH, NF, NT, and SA), we conducted GWAS using a mixed linear model (MLM) and accordingly identified 112 SNPs with significant associations, among which 28, 23, 21, 16, 16, 4, and 4 SNPs were associated with LAD, NT, DIAM, LAL, NF, LENGTH, and SA, respectively, with a threshold of ≥4 − *log_10_*(*p*) (Table 1). The SNPs associated with LAD were distributed on seven chromosomes (Chr. 3, 6, 12, 13, 14, 16X, and 18). Similarly, SNPs associated with NT were distributed on five chromosomes (Chr. 1, 9, 11, 13, and 2). The chromosomal distribution of other SNPs associated with DIAM, LAL, LENGTH, NF, and SA are shown in Table 1. Among the detected SNPs, that with the highest −*log_10_* (*p*) value (5.78) was the LAL-associated AX-90477018 located at 7419551 bp on Chr. 7, followed by SNP AX-90402282 located at 58817483 bp on Chr. 18 with a *−log10* (*p*) value of 5.28.

We also established that 24 of the identified SNPs were co-associated with two of the seven traits. Among the 28 significant SNPs associated with the LAD trait, 17 SNPs present on Chr. 3 (6 SNPs), Chr. 6 (7), Chr. 14 (2), and Chr. 18 (2) were also commonly associated with DIAM. Likewise, three SNPs were identified as commonly associated with LENGTH and NT on Chr. 11 and a further four SNPs distributed on Chr. 8 were found to be commonly associated with NF and SA. Manhattan and quantile-quantile plots of all the significant SNPs associated with individual traits are shown in Figure 2A,B, Figure 3A,B, and Figure 4A–C.

### 2.5. Putative Candidate Genes

Candidate genes were identified within 100-kb regions on either side of 82 significant SNPs, with a total of 570 putative candidate genes being identified for the most significant SNPs associated with the seven root traits, based on MLM modeling (Appendix A). A complete list of all genes associated with significant SNPs is presented in Appendix A. Based on obtained annotation information, possible role in root development, and expression profiles related to root organs/tissues (e.g., roots, root stripped, root nodules, and root tips), we drew up a shortlist of 55 candidate genes (Appendix A). We first used the ePlant (https://bar.utoronto.ca/eplant_soybean/, accessed on 16 April 2022) database to analyze the expression patterns of 55 candidate genes in different tissues. The result showed that all candidate genes were expressed in soybean root tissues (Appendix A). Furthermore, we selected the six highest expressed genes (*Glyma.11g209100*, *Glyma.09g051100, Glyma.01g220600*, *Glyma.05g225700 Glyma.11g209200*, and *Glyma.13g261700*) in the root, and data analysis was completed for these six genes using RNA-Seq soybean libraries (4085) and compared with other tissues (leaf, seedling, shoot, stem, meristem, flower, pod, nodule, seed, embryo, and endosperm). The result showed differential expression levels of six selected candidate genes in the othertissues (Appendix A).

### 2.6. Comparative Genome and Orthologous Gene Analysis

For comparative genomic research to estimate the integrity of the shortlisted candidate genes first, we chose four closely related species, which are listed as follows: soybean, barrel medic (*Medicago truncatula*), adzuki bean (*Vigna augularis*), and common bean (*Phaseolus vulgaris*). A total of 13,191 core gene clusters were found in the four species, and 960 gene clusters were unique to soybeans (Appendix A), with specific gene clusters accounting for ~3.73% (960/22159). Then, we used the root tissue-specific 55 candidate genes for comparative genomic analysis, but did not find any unique gene cluster (Appendix A); however, 41 core gene clusters were found in a comparative analysis of candidate genes and mungbean, adzuki bean, and common bean, with specific core genes clusters accounting for 74.54% (41/55), a result which showed that these candidate genes have common ortholog among interspecies. In Appendix A, 41 core genes are involved in various biological, molecular, and cellular processes. Detailed statistics of shared gene clusters and protein count concerning the cluster are represented in Appendix A.

## 3. Discussion

Although the roots of plants play essential roles in anchoring and the absorption of water and nutrients, their subterranean distribution throughout the plant life cycle makes it difficult to continuously monitor root attributes or traits. Moreover, compared with the traits of other plant parts, relatively less attention has been focused on root genomics. In recent years, however, there has been a gradual increase in the recognition of the importance of root traits, and it is now believed that by optimally exploiting root trait information, we may be on the verge of a “second green revolution” in agriculture [24].

In the present study, we detected significant differences among soybean landraces with respect to seven assessed root traits (DIAM, LAL, LAD, LENGTH, NT, NF, and SA) at (*p* < 0.0001) level (Appendix A). Genetic analysis of the 357 assessed landraces revealed the presence of distinct genotypes among these landraces, with genotypic cluster analysis indicating three major clusters, characterized by varying degrees of admixture (Figure 1A) and a lack of distinct separation based on landrace origin. However, the majority of landraces of Korean origin were found to be divided mainly between two subpopulation groups (I and III). The results of subsequent phylogenetic and PCA analyses consistently revealed the clustering of the assessed landraces into the three major groups indicated by population structure analysis (Figure 1). In our PCA, principal components 1 and 2 were found to account for 34.23% and 14.10% of the observed variance, respectively; however, we detected no distinct separation corresponding to the geographical origin of landraces. These findings tend to be consistent with those of previous genetic studies conducted on soybean, which have reported similar clustering results [25,26,27].

In self-pollinated crops, the rate of LD decay is typically higher than that detected in cross-pollinated plants [28]. Based on physical distance, the LD decay of the entire genome is assumed to increase. In the present study, we obtained an estimated decay of *r^2^* = 0.2 within approximately 200 kb, which is slightly lower than the previously reported LD decay in soybean (235–375 kb) [29,30,31], although it is higher than that reported for other self-pollinated species such as rice and sorghum, with calculated LD decays of (123–167 kb) [32] and (150 kb) [33], respectively. We speculate that these differences could be attributed to a lower genome marker coverage and the smaller number of genotypes in the case of rice and sorghum.

Although numerous studies have examined the agronomic traits of soybean, very few have adopted a GWAS approach to assessing the genetic basis of root traits in soybean. To facilitate such studies, a necessary prerequisite is genetic diversity among the assessed germplasm resources. In terms of allelic diversity for various agronomic traits, the most extensive differences can be found among wild soybeans, followed by landraces, with cultivated soybeans being characterized by being the least diverse. Accordingly, for GWAS in the present study, we used a diverse collection of landraces (*n* = 357) to identify genetic loci associated with root traits. The primary goal of this study was to determine the genetic basis of phenotypic variation in the soybean root system. To this end, we used a 180KSoyaSNP array for our analyses of root traits. In GWAS studies, marker density is of particular importance with respect to establishing entire genome coverage and capturing all the available haplotypic variation, with a larger number of markers being conducive to obtaining sufficient LD to identify significant marker–trait associations. To the best of our knowledge, the marker dataset used in the present study is the second-largest examined to date for root trait GWAS, after that recently reported by Seck et al. [21]. Comparatively, previous studies that have examined root traits based on a combination of QTL mapping and GWAS have tended to use smaller sets of markers ranging in number from 232 to 38,052 [16,22,23,34].

In this study, we identified a total of 112 significant SNPs associated with one or more of the seven assessed root traits, which enabled us to identify genomic regions associated with desired traits. Similarly, recent GWAS studies conducted for root traits in soybean detected 10 genomic regions for the length and diameter of roots and four QTL regions for the number of lateral roots [21,23]. Given the sufficient marker density obtained in the present study, we were able to detect a large number of significant SNPs with a *−log_10_(p)* value >4.0 for the all-measured traits using an MLM model. The phenotypes showed high stability across replicate analyses, and we observed significant genetic diversity among the 357 landraces with respect to the seven assessed root traits. Moreover, we also succeeded in detecting a large number of SNPs associated with the traits of interest. Based on comparisons of the significant SNPs detected in the present study and those previously reported for root and shoot traits, we identified significant SNPs on chromosome 13 that differ from the SNPs previously reported for root traits, although interestingly these SNPs map to locations within the vicinity of those detected for NT (AX-90325580, AX-90370286, AX-90457815, AX-90452276, AX-90480597, AX-90305162, and AX-90433497) [15,35]. In addition, soybean locus ss715616115 on chromosome 13 (position 33415484), previously reported by Zeng et al. (2017) to be associated with salt tolerance, is also located in the vicinity of the aforementioned SNPs. In addition, the ss715624611 locus associated with salt tolerance on chromosome 16 (position 33383414) harbors the SNP AX-90502945 (position 33527990) associated with LAD. Furthermore, common SNPs detected for NF and SA (AX-90428520, AX-90467878, AX-90468823, and AX-90412442) in the present study map within previously identified QTLs associated with different root traits. For example, SNPs associated with root dry weight, taproot length, shoot length, lateral root number, and total root length were detected on chromosome 8 at the seedling stage in root phenotyping studies [22,34,36,37,38]. Similarly, SNPs present on chromosome 6 are commonly associated with DIAM and LAD (AX-90513850, AX-90502675, AX-90329829, AX-90504776, AX-90434642, AX-90416589, and AX-90426968) and SNPs on chromosome 18 associated with DIAM, LAD, and LAL are located in the vicinity of previously reported SNPs for SA, root volume, and branching number in soybean [15], as well as photosynthetic traits associated with phosphorus efficiency [39]. To the best of our knowledge, however, most of the remaining SNPs we identified as being associated with root traits have not been reported previously.

Having identified SNPs associated with the traits of interest, we proceeded to identify the putative candidate genes for the significant SNPs, which are listed in Appendix A. In total, we detected 570 genes within the 100-kb regions up and downstream of the SNPs, which we subsequently whittled down to a shortlist of 55 genes based on annotation information, possible roles in root development, and expression profiles related to root organs/tissues. Recently, Seck et al. (2020), using a similar approach, identified two relevant candidate genes (*Glyma.03g065700* and *Glyma.07g096000*) associated with soybean root development and root tips. Furthermore, six genes that showed higher expression in the root (Appendix A) were compared with other plant tissue to see the expression across tissues, which showed a differential expression pattern. A similar approach was used recently to elucidate the role of candidate genes for tocopherol content in soybean seed [40]. Among the genes, we identified *Glyma.09g073300* and *Glyma.01g220600*, the functional annotations of which in the public domain are auxin-responsive protein-related protein and aquaporin PIP1-4-related genes, respectively. Considering other highly expressed genes in the root would be of particular interest to further validate and assess the expression of these genes in the root tissues of contrasting landraces. Apart from these genes, further research will be necessary to validate and confirm the identity of the remaining genes, particularly those genes for which annotation information is currently unavailable. Moreover, it is desirable to analyze the expression of all the promising candidate genes to determine genotype/population differences, as well as their patterns of expression at different stages of root development.

## 4. Materials and Methods

### 4.1. Plant Materials

In this study, we used 357 soybean landraces collected from the National Agrobiodiversity Center gene bank of the Rural Development Administration. The collected landraces were derived from the following four countries: South Korea (115), North Korea (58), China (90), and Japan (94). The seeds of these germplasms were sown in polyvinyl chloride tubes (8 cm in diameter and 40 cm in height) containing horticultural soil (Tobirang, Baekkwang Fertility, Republic of Korea) in a randomized block design with three replicates. All landraces were grown for 40 days and were harvested at the V2 stage of growth.

### 4.2. Root Phenotype Evaluation

For greenhouse experiments, individual plant samples were removed and measured at the V2 growth stage, and having been rinsed with clean fresh water, their root samples were measured. Seven root traits, namely, average root diameter (DIAM), link average diameter (LAD), link average length (LAL), total root length (LENGTH), number of root forks (NF), number of root tips (NT), and surface area (SA), were scanned and analyzed using the Win-RHIZO system (LA6400XL; Canada Regent Instruments). We had performed this root phenotyping previously [41] and used the phenotypic data for conducting GWAS of the 357 assessed landraces.

### 4.3. Genotyping, Population Structure, and Linkage Disequilibrium Analyses

For DNA extraction, we collected samples of young trifoliate leaves from individual plants of each of the 357 soybean landraces, from which the genomic DNAs were isolated using the magnetic bead method (Qiagen, Hilden, Germany) following the manufacturer’s instructions. Genotyping of the 357 individuals was performed using an Axiom^®^ 180k SoyaSNP array following a previously described method [42]. The genotypic data thus obtained were filtered based on missing IDs/SNPs and a minor allele frequency of ≤0.01, and haplotype phasing and imputation of individuals with a minimum heterozygosity proportion of ≤0.05 were conducted using BEAGLE version 3.3.2 (https://faculty.washington.edu/browning/beagle/beagle.html, accessed on 14 April 2022) based on a hidden Markov process algorithm. The number of subpopulations (*k*) was determined based on cluster analysis, and further ancestry estimation was performed based on ADMIXTURE analysis [43]. A bar plot of the 357 individuals was generated using STRUCTURE 2.3.4 [44,45] for a *k* = value of 3. Linkage disequilibrium (LD) analysis was performed using genome-wide SNP markers, and correlation (*r*^2^) values among the 84,210 SNP markers were calculated using QTLmaxV2 software based on PLINK, with a window size of 400 SNP markers, and estimated as the squared values of the correlation coefficient between SNP marker pairs [46].

### 4.4. Phylogenetic and Principal Component Analyses 

Phylogenetic analysis was performed using the similarity identity-by-state coefficient matrix calculated using Plink v2.0 (www.cog-genomics.org/plink/2.0/, accessed on 14 April 2022) [47], and a phylogenetic tree was constructed based on the genetic similarity matrix unweighted paired group method using arithmetic averages (UPGMA). Principal component analysis (PCA) was performed using the R package ggplot2 distance matrix.

### 4.5. Genome-Wide Association Studies 

All GWAS analyses were performed for seven root traits of the 357 landraces using GWASpro (QTLmaxV2) [48] with a linear mixed model and default settings. We filtered the 180,375 SNPs based on the previously mentioned parameters, and the 84,210 SNP markers thus obtained were used for GWAS analysis. We set the threshold for the *−log10(p)* values ≥ 4 at alpha = 5 to search for significant SNPs.

### 4.6. Candidate Gene Identification

We used the most significant SNPs detected in GWAS to identify potential candidate genes, based on annotation using the SoyBase database (https://soybase.org/SequenceIntro.php, accessed on15 April 2022) according to the Wm82. a2. v1 soybean reference genome. Candidate genes were searched within a sequence range of 100 kb up and downstream each significant SNP using the genome browser, and functional annotation of the genes was performed using the Phytozome database [49].

### 4.7. RNA-Seq Data Analysis of Selected Candidate Genes

To determine the candidate genes expression level, we first performed an analysis using the ePlant (https://bar.utoronto.ca/eplant_soybean/, accessed on 4 December 2022) database and a heatmap was constructed for shortlisted genes using TBtools software (https://github.com/CJ-Chen/Tbtools, accessed on 4 December 2022). Furthermore, selected genes were used for differential expression analysis at different tissues using web-based publicly available RNA-Seq soybean libraries’ (4085) data with default setting (http://ipf.sustech.edu.cn/pub/soybean/, accessed on 4 December 2022) [50]. Furthermore, the whole genome and candidate genes among closely related legume species were compared and orthologous gene clusters were determined using OrthoVenn2 web portal [51].

### 4.8. Statistical Analysis

We used phenotypic data for the 357 landraces obtained in our recently published study [41]. To determine statistical significance, descriptive statistics were obtained for the landrace data using SAS release 9.4 (SAS, Gary, NC, USA), and a histogram was generated for the seven assessed root traits.

## 5. Conclusions

In this study, using GWAS, we determined the genomic regions associated with seven selected root traits in soybean landraces and identified the 112 most significant SNPs associated with the measured traits. Furthermore, based on annotation information and publicly available expression data, putative candidate genes were identified in the vicinities of the SNPs. These promising candidate genes could potentially serve as molecular targets for further studies conducted to evaluate their role in the regulation of root development. Moreover, the numerous SNPs identified in this study can be used as a potential resource in soybean breeding programs, with a view toward marker development for root traits. Collectively, our findings will provide a basis for identifying the genetic components underlying phenotypic variation in the root traits of soybean landraces.

## Figures and Tables

**Figure 1 ijms-24-00873-f001:**
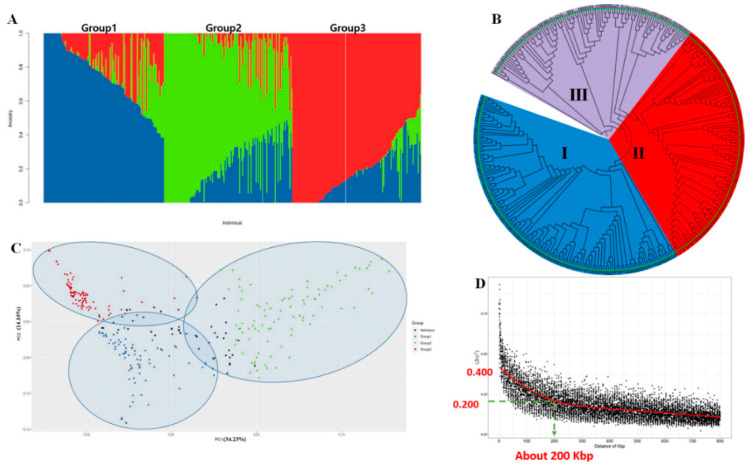
Population structure of 357 soybean landraces. (**A**) A bar plot diagram showing the results of clustering analysis when the number of subgroups (K) = 3. The colors blue, green, and red represent separate groups with different levels of admixture. (**B**) A phylogenetic tree of 357 soybean landraces. (**C**) Principal component plot for the 357 soybean landraces. (**D**) Genome-wide linkage disequilibrium (LD) decay for all 357 soybean landraces. *R*^2^ indicates the squared allele frequency correlations between all pairs of SNP markers. The *X*-axis indicates the distance between marker pairs.

**Figure 2 ijms-24-00873-f002:**
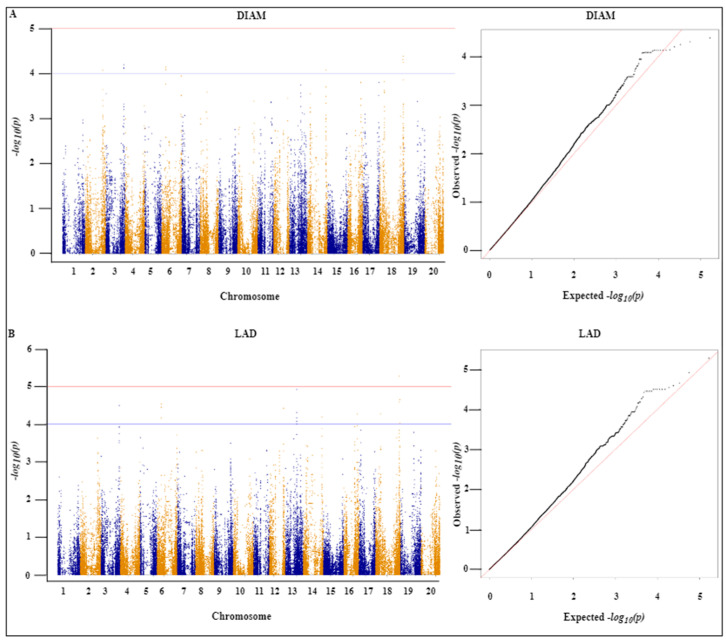
Manhattan and quantile-quantile (QQ) plots of the 357 soybean landraces using the MLM model. (**A**) Root average diameter (DIAM) and (**B**) link average diameter (LAD). The *X*-axis represents the chromosome number, and the Y-axis represents the −*log_10_p*) value. The threshold of 4.0 (Bonferroni correction) was adopted with the blue line in the Manhattan plots. The Manhattan and QQ plots are based on analyses of the association between 84,210 chromosomal SNPs and DIAM and LAD traits.

**Figure 3 ijms-24-00873-f003:**
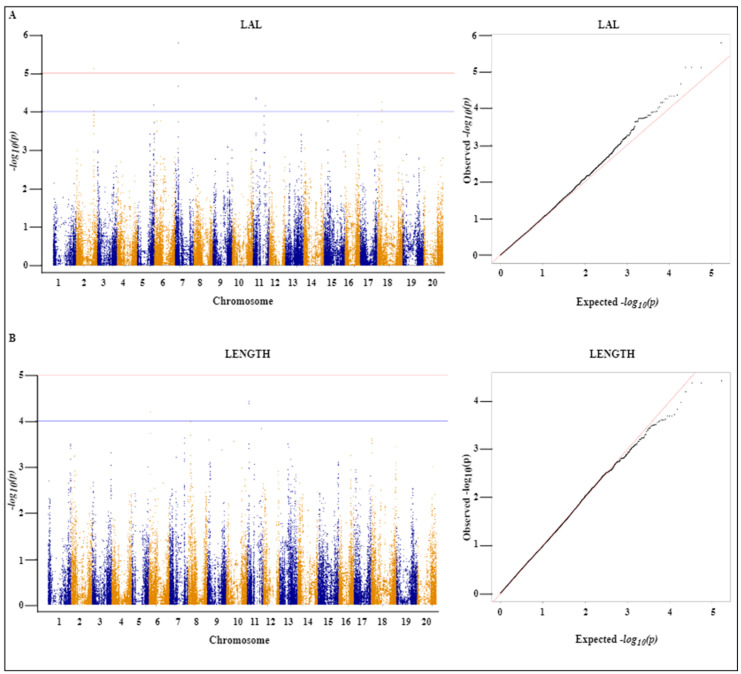
Manhattan and quantile-quantile (QQ) plots of the 357 soybean landraces using the MLM model. (**A**) Link average length (LAL) and (**B**) total root length (LENGTH). The X-axis represents the chromosome number and the Y-axis represents the *−log10*(*p*) value. The threshold of 4.0 (Bonferroni correction) was adopted with the blue line in the Manhattan plots. The Manhattan and QQ plots are based on analyses of the associations between 84,210 chromosomal SNPs and DIAM and LAD traits.

**Figure 4 ijms-24-00873-f004:**
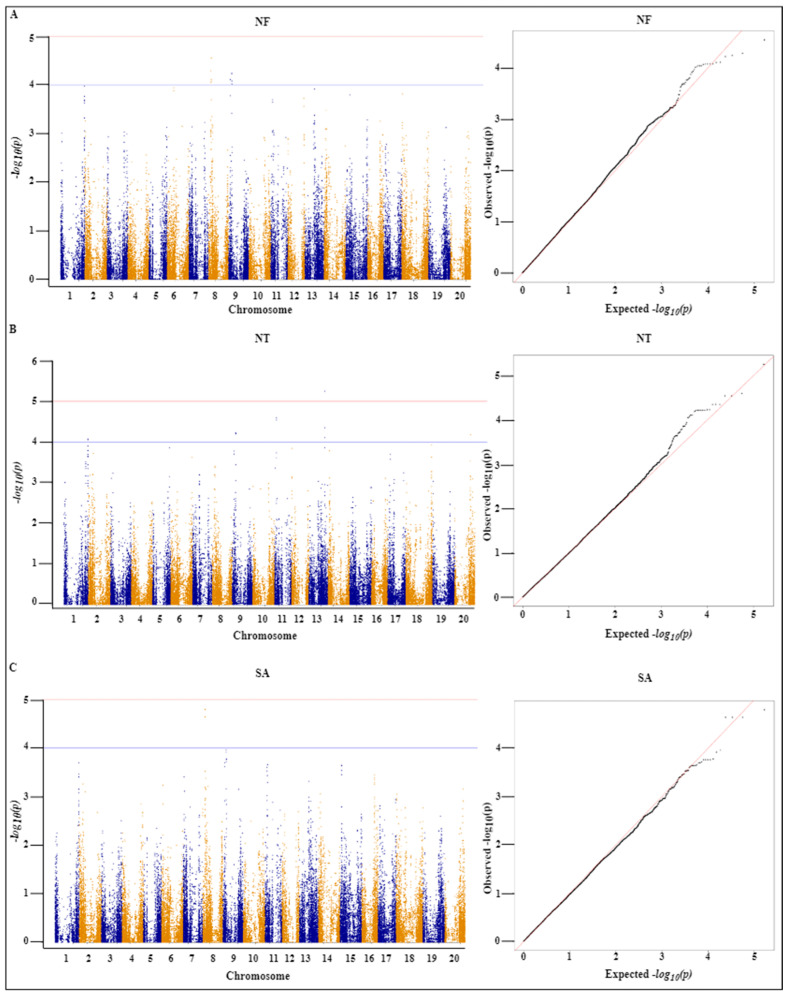
Manhattan and quantile-quantile (QQ) plots of the 357 soybean landraces using the MLM model. (**A**) Number of root forks (NF), (**B**) number of root tips (NT), and (**C**) root surface area (SA). The X-axis represents the chromosome number and the Y-axis represents the *−log10*(*p*) value. The threshold of 4.0 (Bonferroni correction) was adopted with the blue line in the Manhattan plots. The Manhattan plot and QQ plot is based on the association of 84,210 chromosomal SNPs with DIAM and LAD traits.

**Table 1 ijms-24-00873-t001:** Significant SNPs identified for root traits using the MLM model.

Trait	SNP	Chr	Position	Ref	Alt	*p* Value	−log10(*p*)
DIAM	AX−90460045	2	43709318	G	A	8.60 × 10^−5^	4.07
	AX−90505093	3	44598254	C	G	6.41 × 10^−5^	4.19
	AX−90311611	3	44782942	G	A	7.53 × 10^−5^	4.12
	AX−90308307	3	44785822	G	A	7.53 × 10^−5^	4.12
	AX−90385325	3	44786068	A	G	7.53 × 10^−5^	4.12
	AX−90442177	3	44786975	T	C	7.53 × 10^−5^	4.12
	AX−90503377	3	44787733	A	G	7.53 × 10^−5^	4.12
	AX−90345460	3	44799953	T	A	7.53 × 10^−5^	4.12
	AX−90385554	3	44815582	C	T	7.76 × 10^−5^	4.11
	AX−90513850	6	10393414	G	A	8.32 × 10^−5^	4.08
	AX−90502675	6	10469343	G	A	8.32 × 10^−5^	4.08
	AX−90329829	6	10471655	A	G	7.33 × 10^−5^	4.14
	AX−90504776	6	10472539	T	A	8.32 × 10^−5^	4.08
	AX−90434642	6	10478559	A	G	8.32 × 10^−5^	4.08
	AX−90416589	6	10495842	A	G	8.32 × 10^−5^	4.08
	AX−90426968	6	10501950	A	G	8.32 × 10^−5^	4.08
	AX−90524509	14	46332487	T	C	8.41 × 10^−5^	4.08
	AX−90332250	14	46340213	G	T	8.41 × 10^−5^	4.08
	AX−90367887	18	58632433	T	G	5.02 × 10^−5^	4.3
	AX−90402282	18	58817483	A	G	4.19 × 10^−5^	4.38
	AX−90468047	18	59089827	A	C	5.72 × 10^−5^	4.24
LAD	AX−90311611	3	44782942	G	A	3.18 × 10^−5^	4.5
	AX−90308307	3	44785822	G	A	3.18 × 10^−5^	4.5
	AX−90385325	3	44786068	A	G	3.18 × 10^−5^	4.5
	AX−90442177	3	44786975	T	C	3.18 × 10^−5^	4.5
	AX−90503377	3	44787733	A	G	3.18 × 10^−5^	4.5
	AX−90345460	3	44799953	T	A	3.18 × 10^−5^	4.5
	AX−90513850	6	10393414	G	A	3.53 × 10^−5^	4.45
	AX−90454547	6	10396428	A	C	6.94 × 10^−5^	4.16
	AX−90502675	6	10469343	G	A	3.53 × 10^−5^	4.45
	AX−90329829	6	10471655	A	G	2.86 × 10^−5^	4.54
	AX−90504776	6	10472539	T	A	3.53 × 10^−5^	4.45
	AX−90434642	6	10478559	A	G	3.53 × 10^−5^	4.45
	AX−90416589	6	10495842	A	G	3.53 × 10^−5^	4.45
	AX−90426968	6	10501950	A	G	3.53 × 10^−5^	4.45
	AX−90481233	12	35529714	A	G	3.77 × 10^−5^	4.42
	AX−90405818	13	27859216	T	C	4.91 × 10^−5^	4.31
	AX−90405799	13	27870065	G	A	1.21 × 10^−5^	4.92
	AX−90496747	13	27877321	G	A	8.44 × 10^−5^	4.07
	AX−90521967	13	27901991	T	G	6.88 × 10^−5^	4.16
	AX−90336511	13	27939975	C	T	9.79 × 10^−5^	4.01
	AX−90524509	14	46332487	T	C	6.53 × 10^−5^	4.18
	AX−90332250	14	46340213	G	T	6.53 × 10^−5^	4.18
	AX−90502945	16	33527990	A	G	5.38 × 10^−5^	4.27
	AX−90422071	18	12128963	G	T	5.35 × 10^−5^	4.27
	AX−90367887	18	58632433	G	A	2.55 × 10^−5^	4.59
	AX−90402282	18	58817483	A	C	5.28 × 10^−6^	5.28
	AX−90365160	18	60026624	G	A	2.23 × 10^−5^	4.65
	AX−90495375	18	60153725	A	C	9.16 × 10^−5^	4.04
LAL	AX−90523253	2	42676896	A	C	9.88 × 10^−5^	4.01
	AX−90367890	2	42742102	T	G	7.71 × 10^−6^	5.11
	AX−90407903	2	42742896	G	A	7.71 × 10^−6^	5.11
	AX−90372917	2	42754007	T	C	7.71 × 10^−6^	5.11
	AX−90456562	5	40343307	C	A	6.75 × 10^−5^	4.17
	AX−90365252	7	7177786	A	G	2.20 × 10^−5^	4.66
	AX−90477018	7	7419551	A	G	1.65 × 10^−6^	5.78
	AX−90491184	11	7212890	G	T	4.76 × 10^−5^	4.32
	AX−90348822	11	7234607	T	G	4.76 × 10^−5^	4.32
	AX−90337775	11	7234726	T	G	4.36 × 10^−5^	4.36
	AX−90498877	11	7234867	A	G	4.76 × 10^−5^	4.32
	AX−90443985	11	30156429	C	T	7.06 × 10^−5^	4.15
	AX−90340335	11	30159162	A	G	7.06 × 10^−5^	4.15
	AX−90362827	18	11624535	G	T	9.16 × 10^−5^	4.04
	AX−90513611	18	11665000	T	C	5.60 × 10^−5^	4.25
	AX−90396575	18	11670146	G	T	5.60 × 10^−5^	4.25
LENGTH	AX−90431861	6	3169898	G	A	6.34 × 10^−5^	4.2
	AX−90446460	11	4302823	C	T	4.20 × 10^−5^	4.38
	AX−90414551	11	4333105	C	G	4.20 × 10^−5^	4.38
	AX−90472866	11	4337689	T	G	3.75 × 10^−5^	4.43
NF	AX−90428520	8	6172141	G	A	5.20 × 10^−5^	4.28
	AX−90467878	8	6174931	T	C	9.11 × 10^−5^	4.04
	AX−90468823	8	6176238	C	T	9.11 × 10^−5^	4.04
	AX−90412442	8	6176371	A	G	9.11 × 10^−5^	4.04
	AX−90493535	8	7168291	C	A	8.54 × 10^−5^	4.07
	AX−90455156	8	7168816	G	T	7.81 × 10^−5^	4.11
	AX−90331992	8	7170112	C	A	2.84 × 10^−5^	4.55
	AX−90365804	9	4467094	T	C	7.79 × 10^−5^	4.11
	AX−90495283	9	7691924	A	C	9.53 × 10^−5^	4.02
	AX−90449059	9	7699360	G	T	5.73 × 10^−5^	4.24
	AX−90520390	9	7709441	T	G	8.40 × 10^−5^	4.08
	AX−90434197	9	7713796	G	A	8.40 × 10^−5^	4.08
	AX−90498061	9	7719800	A	G	6.01 × 10^−5^	4.22
	AX−90419697	9	7726552	G	A	8.40 × 10^−5^	4.08
	AX−90492156	9	7734112	A	G	8.40 × 10^−5^	4.08
	AX−90493039	9	7740050	A	G	9.67 × 10^−5^	4.01
NT	AX−90522644	1	54922179	G	T	8.83 × 10^−5^	4.05
	AX−90466067	1	54927068	A	C	8.36 × 10^−5^	4.08
	AX−90467825	1	54937834	T	C	8.83 × 10^−5^	4.05
	AX−90430136	9	7783387	A	G	6.06 × 10^−5^	4.22
	AX−90419830	9	7795507	G	T	6.06 × 10^−5^	4.22
	AX−90317331	9	7816745	A	G	5.91 × 10^−5^	4.23
	AX−90313864	9	7826748	C	T	5.91 × 10^−5^	4.23
	AX−90431604	9	7832817	A	G	6.32 × 10^−5^	4.2
	AX−90382601	9	7840359	G	A	6.06 × 10^−5^	4.22
	AX−90312113	9	7847844	T	G	6.06 × 10^−5^	4.22
	AX−90455909	9	7848294	A	C	6.06 × 10^−5^	4.22
	AX−90306078	9	7849987	T	C	6.06 × 10^−5^	4.22
	AX−90446460	11	4302823	C	T	2.85 × 10^−5^	4.54
	AX−90414551	11	4333105	C	G	2.85 × 10^−5^	4.54
	AX−90472866	11	4337689	T	G	2.52 × 10^−5^	4.6
	AX−90325580	13	36648560	A	T	4.45 × 10^−5^	4.35
	AX−90370286	13	36650121	A	G	4.45 × 10^−5^	4.35
	AX−90457815	13	36655953	G	A	5.60 × 10^−6^	5.25
	AX−90452276	13	36656131	G	T	4.45 × 10^−5^	4.35
	AX−90480597	13	36667146	G	A	7.85 × 10^−5^	4.11
	AX−90305162	13	36668072	A	G	7.85 × 10^−5^	4.11
	AX−90433497	13	36669907	G	A	7.85 × 10^−5^	4.11
	AX−90383650	20	36657615	G	T	6.56 × 10^−5^	4.18
SA	AX−90428520	8	6172141	G	A	1.61 × 10^−5^	4.79
	AX−90467878	8	6174931	T	C	2.30 × 10^−5^	4.64
	AX−90468823	8	6176238	C	T	2.30 × 10^−5^	4.64
	AX−90412442	8	6176371	A	G	2.30 × 10^−5^	4.64

REF, Reference allele based on the Williams 82 reference genome version Wm82. a2. v1; ALT, Alternative allele.

## Data Availability

Not applicable.

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
