# Peer review of "Genome-Wide Association Studies of Seven Root Traits in Soybean (Glycine max L.) Landraces"

_ijms, 2023, doi:10.3390/ijms24010873_

Round 1

Reviewer 1 Report

file attached

Author Response

Author's Reply to the Review Report (Reviewer 1)

Answer: We would like to thank the worthy reviewer for the time given to this manuscript. All the comments and suggestions were genuine, valuable and improved the quality of the manuscript. We highly appreciate your efforts and agree to the suggested changes. All the changes that were made in the revised MS can be found with the track change, highlighted in green color.

Please provide the expanded form of abbreviated words and acronyms when using the first time.

Answer: Thank you for the suggestion, we have provided an expanded form of abbreviated words and acronyms at necessary places.

The introduction section can be shortened to focus on the questions beings addressed in this study along with the details of earlier studies and significance.

Answer: Thank you for the suggestion, as per the suggestions we have incorporated changes.

It would be informative if the authors have included diverse locations also.

Answer: Thank you for the suggestion, We agree that a diverse location would have been more informative, and able to find more stable quantitative trait nucleotides (QTNs), but initially we aimed to examine the phenotypic association among root traits, elucidate their genetic basis, and identify significant SNPs associated with regions/loci controlling root traits. We will investigate the genotype’s diverse locations in the next step of the experiment.

Authors can add utility of using GWAS tools and their efficiency in the identification of significant genes/QTL’s.

Answer: Thank you for the suggestion, we have mentioned the utility of GWAS tools and their efficiency in the identification of significant genes/QTLs,  in the introduction, Lines 85-90.

The author can add the SNP variation (transition/transversion) for each gene associated with the trait.

Answer: Thank you for the suggestion, We have put the reference and alternate allele (SNPs) information in the Table 1. we have a further plan to conduct the expression analysis of selected genes in root and other plant tissue and then look for only those genes which show the correlation with phenotypes and then add the SNP (transition/transversion) variation for specific genes.

Reviewer 2 Report

In this study, the authors described the GWAS results of seven root traits in a soybean landrace population with 357 samples. The authors tabulated candidate QTLs they found and discussed several candidate genes in the QTL regions. These results may be of some value for soybean research and breeding. The research design and experimental scheme are reasonable on the whole, and the results are clear.

However, the phenotype data used in this study has been published by the authors previously and the genotype data was obtained from a 180-K SNP array. There is no doubt that studying soybean root traits is of great significance to plant biology and soybean breeding, but the novelty of data and results in this study is limited.

Here are my comments to the contents of this manuscript:

(1)  Does the population structure show correlation with the seven root traits? The author performed GWAS using MLM, was the principal component matrix considered in this process? The author should provide more details for this analysis.

(2)  “A significant cut-off P-value was calculated using the Bonferroni equation with cut-off 5 (Main) and Suggestive line 4.” This description is not proper, 5 is -log10P value, not P-value itself. In addition, how many independent markers are in the 84,210 markers used for GWAS? According to Bonferroni equation, a test of 84,210 markers doesn’t yield a -log10P cut-off of 5 or 4.

(3)  In population structure analysis, the authors determined the proper number of clusters using K-means and elbow method, please provide the relevant data and plot of this analysis.

(4)  The discussion of candidate genes is too brief. If transcriptomic or other omics data can be used to explore the candidate genes, it may be helpful to improve the quality of the article.

Author Response

Author's Reply to the Review Report (Reviewer 2)

Comments and Suggestions for Authors

In this study, the authors described the GWAS results of seven root traits in a soybean landrace population with 357 samples. The authors tabulated candidate QTLs they found and discussed several candidate genes in the QTL regions. These results may be of some value for soybean research and breeding. The research design and experimental scheme are reasonable on the whole, and the results are clear.

However, the phenotype data used in this study has been published by the authors previously and the genotype data was obtained from a 180-K SNP array. There is no doubt that studying soybean root traits is of great significance to plant biology and soybean breeding, but the novelty of data and results in this study is limited.

Answer: We would like to thank the worthy reviewer for the time given to this manuscript. All the comments and suggestions were genuine, valuable and helped to improve the quality of the manuscript. We highly appreciate your efforts and agree to the suggested changes. All the changes that were made in the revised MS can be found with the track change, highlighted in green color.

Here are my comments to the contents of this manuscript:

(1)  Does the population structure show correlation with the seven root traits? The author performed GWAS using MLM, was the principal component matrix considered in this process? The author should provide more details for this analysis.

Answer: Thank you for commenting on it, yes, the population structure showed a correlation with the analyzed root traits and principal component analysis was also considered. We have explained in the results section as per the suggestions we have provided supplementary Table S2 (Eigenvalue and proportion of principal components). Secondly, as we have already published phenotype data where we have provided some of the analysis previously, it’s not ideal to show the results again we have not represented in this MS. However, we have provided the description “Similarly, PCA-based clustering revealed that the 357 soybean lines can be divided into three major groups, among which there was a certain degree of overlap comprising areas of admixture (Figure 1A). PC1 and PC2 accounted for 34.23% and 14.10% of the observed variation, respectively (Figure 1C). The first three PC1, PC2 and PC3 contributed to the 58.20% variation (Table S2). Based on these analyses, we can thus conclude that the 357 assessed soybean landraces can be divided into three major groups/subpopulations, which represent an admixture of three ancestral populations”.

(2)  “A significant cut-off P-value was calculated using the Bonferroni equation with cut-off 5 (Main) and Suggestive line 4.” This description is not proper, 5 is -log10P value, not P-value itself. In addition, how many independent markers are in the 84,210 markers used for GWAS? According to Bonferroni equation, a test of 84,210 markers doesn’t yield a -log10P cut-off of 5 or 4.

Answer: We modified the sentence thank you for pointing out this error. Instate of P value now we have mentioned the -log10(P) values >4.00 at alpha=5.

We used the GWASpro which introduces a genomic correction function, which uses the genomic inflation factor (λGC). Secondly, we use filtered 84,210 SNPs to calculate the threshold. The Bonferroni-corrected threshold for the -log10(P) values were 6.2 at alpha=0.05, 4.93 at alpha=1, and 4.23 at alpha=5. so, we set the threshold for the -log10(P) values >4.00 at alpha=5 to search for significant SNPs.

(3)  In population structure analysis, the authors determined the proper number of clusters using K-means and elbow method, please provide the relevant data and plot of this analysis.

Answer: We have provided this data in a supplementary file. (Figure S1 and Table S2).

(4)  The discussion of candidate genes is too brief. If transcriptomic or other omics data can be used to explore the candidate genes, it may be helpful to improve the quality of the article.

Answer: We highly appreciate the valuable suggestion and added RNA-Seq soybean libraries (4,085) based data analysis in addition comparative genome and orthologous gene cluster analysis results were added to improve the quality of MS. Information provided in the supplementary file.

Round 2

Reviewer 2 Report

After a new round of revisions, the authors basically answered the questions we were concerned about. I think the current version is acceptable.